# Explaining Multimodal LLMs via Intra-Modal Token Interactions

## Abstract

Multimodal Large Language Models (MLLMs) have achieved remarkable success across diverse vision-language tasks, yet their internal decision-making mechanisms remain insufficiently understood. Existing interpretability research has primarily focused on cross-modal attribution, identifying which image regions the model attends to during output generation. However, these approaches often overlook intra-modal dependencies. In the visual modality, attributing importance to isolated image patches ignores spatial context due to limited receptive fields, resulting in fragmented and noisy explanations. In the textual modality, reliance on preceding tokens introduces spurious activations. Failing to effectively mitigate these interference compromises attribution fidelity. To address these limitations, we propose enhancing interpretability by leveraging intra-modal interaction. For the visual branch, we introduce *Multi-Scale Explanation Aggregation* (MSEA), which aggregates attributions over multi-scale inputs to dynamically adjust receptive fields, producing more holistic and spatially coherent visual explanations. For the textual branch, we propose *Activation Ranking Correlation* (ARC), which measures the relevance of contextual tokens to the current token via alignment of their top-$k$ prediction rankings. ARC leverages this relevance to suppress spurious activations from irrelevant contexts while preserving semantically coherent ones. Extensive experiments across state-of-the-art MLLMs and benchmark datasets demonstrate that our approach consistently outperforms existing interpretability methods, yielding more faithful and fine-grained explanations of model behavior.

## 1 Introduction

Multimodal Large Language Models (MLLMs) Wang et al. (2024); Chen et al. (2024) have demonstrated remarkable performance on a wide range of vision-language tasks, from visual question answering Antol et al. (2015) to image captioning Chen et al. (2015). Despite this progress, the internal reasoning of these models remains a black box. We can easily observe their outputs, but we lack a clear understanding of how or why they arrive at specific conclusions. The lack of transparency not only hinders their deployment in high-stakes applications requiring trust and accountability, but also limits our ability to diagnose errors and systematically improve model design.

To bridge this gap, prior studies on MLLM interpretability Zhang et al. (2025); Ben Melech Stan et al. (2024) have primarily concentrated on cross-modal attribution, aiming to identify the image regions that drive the generation of specific textual responses. Early works adapt conventional gradient-based methods, such as Grad-CAM Selvaraju et al. (2017), or attention-based techniques Lapuschkin et al. (2019), to MLLMs for attribution. More recent approaches Li et al. (2025a); Jiang et al. (2024) leverage the logit lens nostalgebraist (2020), which provides token-level visual attribution by decoding hidden states of visual tokens through the final unembedding layers. This process yields token-generation probabilities, from which attribution maps are derived. While these methods effectively capture inter-modal interactions, they place less emphasis on intra-modal dynamics. For instance, the joint influence of spatially adjacent image patches on visual attribution has not been explicitly modeled. Furthermore, although prior work has highlighted contextual interference introduced by preceding text tokens Li et al. (2025a), strategies for effectively mitigating such effects need further exploration.

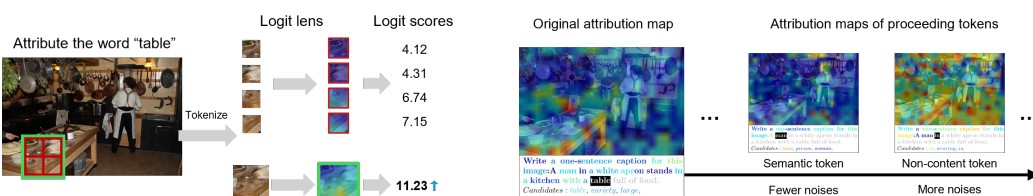

(a) Attribution varies with context window.   (b) Attributions on non-content tokens exhibit higher noise.

Figure 1: Motivation of our proposed MSEA (a) and SAC (b).

We argue that incorporating intra-modal interactions is essential for faithful visual attribution. First, a single visual token corresponds to only a small patch of the input image and thus carries limited contextual information. Decoding the hidden state of such a patch may map to irrelevant words rather than the expect one, leading to misleading attribution. Second, prior studies Li et al. (2025a) suggest that the generation of each token depends on preceding tokens, which results in the activations of current tokens carrying irrelevant information from earlier tokens. If this interference from previous tokens is not properly mitigated, the resulting attribution may be contaminated by noise.

To address these limitations, we propose a new interpretability framework that strengthens intra-modal interactions to produce more faithful and holistic explanations. Our approach comprises two complementary components. For vision, we introduce Multi-Scale Explanation Aggregation (MSEA), which computes and integrates attributions across input contexts of different scales. We feed the model with images at multiple resolutions, where each visual token corresponds to a different receptive field. The visual activations are then decoded using the logit lens nostalgebraist (2020), and the resulting attribution maps are rescaled and fused to form a comprehensive explanation map that aggregates contextual information across scales, thereby improving attribution fidelity. For language, we propose Activation Ranking Correlation (ARC), a method designed to identify and suppress the influence of irrelevant preceding tokens. Specifically, ARC quantifies the relevance of each contextual token to the current decoding step by comparing the ranking order of their top-k predicted tokens with that of the current token. Tokens whose top-k rankings consistently align with the current token are considered semantically coherent and are down-weighted in the mitigation process. In contrast, tokens that induce divergent rankings are regarded as spurious, and their activations are explicitly subtracted from the current token's activation to reduce interference.

We evaluate our framework on several SoTA MLLMs, including LLaVA-1.5 Liu et al. (2024), Qwen2-VL Wang et al. (2024), and InternVL2.5 Chen et al. (2024), using diverse datasets such as COCO Caption Chen et al. (2015), OpenPSG Zhou et al. (2024), and GranDf Rasheed et al. (2024) dataset. To assess scalability, we further test models of varying sizes, ranging from 2B to 13B parameters. Our approach consistently outperforms existing attribution methods, yielding improvements of 3.69%–6.37% on the Qwen2-VL-2B model across different datasets, and 4.98%–14.52% across models of different architectures and scales on the COCO Caption dataset.

Our contributions can be summarized as follows:

- We focus on overlooked intra-modal interactions in MLLM explanations. Specifically, we identify neglected spatial dependencies among visual token attributions and insufficient mitigation of semantic interference among preceding textual tokens, which often lead to unfaithful or noisy explanations.

- We propose a new method to enhance intra-modal attribution fidelity by introducing MSEA, which integrates attributions across multiple image scales to capture spatial context among visual tokens, and ARC, which identifies irrelevant preceding tokens through token-prediction ranking alignment and subtracts their influence to mitigate noise.

- We conduct comprehensive empirical validation on multiple MLLMs (*e.g.*, LLaVA-1.5, Qwen2-VL) across diverse benchmarks and model scales, consistently outperforming existing explainability methods with quantitative improvements ranging from 3.69% to 14.52%.

## 2 RELATED WORK

**Multimodal Large Language Models.** The rapid advancement of computational infrastructure and the availability of large-scale multimodal data have spurred the development of Multimodal Large Language Models (MLLMs), which are capable of perceiving, reasoning over, and generating responses grounded in heterogeneous inputs such as images and text. MLLMs typically leverage pretrained autoregressive language models (Together.xyz, 2023; MosaicML, 2023) as decoders, enabling rapid adaptation to a wide range of vision-language tasks (Antol et al., 2015; Hossain et al., 2019). A common design principle involves introducing a trainable connector module to bridge a pretrained visual encoder with a language model. For example, LLaVA Liu et al. (2024) uses a simple linear projection to map visual features into the LLM's embedding space; Flamingo (Alayrac et al., 2022) employs gated cross-attention to dynamically integrate visual tokens into the language stream. More recent models further enhance alignment: Qwen2-VL Wang et al. (2024) adopts position-aware, dynamic-resolution visual connectors, while InternVL Chen et al. (2024) employs lightweight yet effective MLP-based modules. Despite their impressive performance and emergent reasoning capabilities, the internal decision-making processes of these models remain opaque, highlighting the need for faithful interpretability techniques.

**Visual Attribution for MLLMs.** Visual attribution in Multimodal Large Language Models (MLLMs) has recently drawn increasing attention as a means to interpret how image regions influence textual outputs. Early efforts largely adapt explanation techniques developed for unimodal vision models to the MLLM setting. Representative approaches include gradient-based methods such as CAM Zhou et al. (2016), Grad-CAM Selvaraju et al. (2017), and Grad-CAM++ Chattopadhay et al. (2018), as well as attention-based techniques like LRP Lapuschkin et al. (2019) and Rollout Abnar & Zuidema (2020). These methods typically treat the visual encoder as a standalone module and apply gradient- or attention-based attribution to identify image regions associated with the final prediction. However, such adaptations overlook the autoregressive nature of MLLM generation, where each output token is conditioned on both visual inputs and previously generated text. More recent works address this by enabling token-level visual attribution. For instance, LLaVA-CAM Zhang et al. (2025) and LVLM-Interpret Ben Melech Stan et al. (2024) leverage attention flows or LRP to link image regions to specific output tokens. PROJECTAWAY Jiang et al. (2024) and TAM Li et al. (2025a) adopt the logit lens nostalgebraist (2020) framework to decode visual token activations into word probabilities for attribution. Notably, TAM Li et al. (2025a) first observes that earlier context tokens can introduce redundant visual activations that interfere with later token predictions, and proposes an interference mitigation strategy. Despite these advances, existing methods predominantly focus on inter-modal interactions, while largely neglecting intra-modal dynamics. Specifically, they fail to model spatial context among neighboring visual tokens and lack effective methods to suppress semantic interference from preceding textual tokens. Both limitations can significantly distort attribution fidelity, a gap our work aims to address these issues.

## 3 METHOD

### 3.1 PRELIMINARY

**Multimodal Large Language Models (MLLMs)** Wang et al. (2024); Chen et al. (2024) integrate visual and textual inputs to generate coherent language outputs. Given an image $I$ and a sequence of proceeding text tokens $T_{<t} = \{T_1, \ldots, T_{t-1}\}$, an MLLM predicts the next token $T_t$ by jointly modeling both modalities through a stack of $L$ transformer layers. The image is first tokenized into visual tokens $\mathbf{V} = \{v_1, \ldots, v_N\}$, which are embedded as visual embeddings $\mathbf{E}_v$, while textual tokens are mapped to text embeddings $\mathbf{E}_t$. These embeddings are concatenated and fed into the multimodal transformer to produce contextualized hidden states. To distinguish modalities, the hidden state of a visual token $v_i$ at the $l^{\text{th}}$ layer is denoted by $\mathbf{z}_i^l$, whereas the hidden state of a text token $T_t$ is denoted by $\mathbf{h}_t^L$. At the final layer, the hidden state of the current text token $\mathbf{h}_t^L$ is projected onto the vocabulary space through the unembedding matrix $\mathbf{W}_U$, yielding the logits. The next-token probability distribution is then given by

$$P(T_t \mid T_{<t}, I) = \text{softmax}(\mathbf{W}_U \mathbf{h}_t^L), \tag{1}$$

where $\mathbf{h}_t^L$ denotes the hidden state of the $t$-th text tokens at the $L^{\text{th}}$ layer.

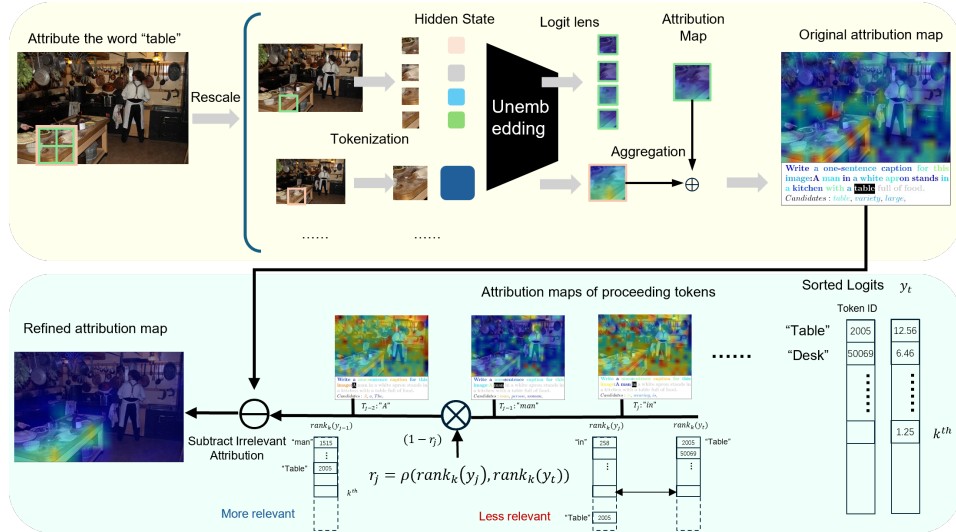

Figure 2: Overview of our proposed framework.

**Logit lens** nostalgebraist (2020) provides a mechanism to interpret how hidden states of visual tokens implicitly contribute to text generation. Logit lens directly projects the hidden states of visual tokens onto the vocabulary space, revealing their latent alignment with textual outputs. Let $\mathcal{V}$ denote the model vocabulary, and consider a generated text token $T_t$ to be explained. For each visual token $v_i$ with hidden state $\mathbf{z}_i^l$ at the $l^{\text{th}}$ layer ($l \leq L$), we compute its contribution to the target text token $T_t$ corresponding to vocabulary index $k$ (i.e., $\mathcal{V}[k] = T_t$) as the logit score:

$$a_i^l(k) = \left[\mathbf{W}_U \mathbf{z}_i^l\right]_k, \tag{2}$$

where $[\cdot]_k$ denotes selecting the $k$-th element of the logits vector. This score directly quantifies the influence of the visual token $v_i$ on generating the specific textual token $T_t$. By aggregating the scores $a_i^l(k)$ across all visual tokens, we construct a visual attribution map that highlights the image regions most responsible for the generation of $T_t$. This formulation provides a direct and interpretable way to attribute textual outputs to specific visual inputs.

### 3.2 MULTI-SCALE EXPLANATION AGGREGATION (MSEA)

Existing logit-lens attribution methods operate at the token level, projecting isolated tokens into the vocabulary space. However, each visual token typically corresponds to only a small region of the image, often covering just part of an object, and therefore carries limited contextual information. As demonstrated in Figure 1a, decoding such hidden states may yield low activation on explain token that do not faithfully reflect the underlying visual evidence. A natural question is whether we can directly incorporate the hidden states of adjacent visual tokens to capture richer context. However, due to the black-box nature of MLLMs, manipulating internal feature interactions is highly challenging, and it is difficult to validate whether such manipulations are semantically faithful.

We propose a simple and effective strategy that operates at the input–output level: aggregating visual attribution outputs across inputs of multiple scales. By feeding images of different resolutions into the model, the receptive fields of visual tokens are varied. This allows us to examine whether new hidden states project onto logits with high scores for the target semantic token (indicating additional semantic content) or remain inactive (indicating no relevant content). This motivates our method, *Multi-Scale Explanation Aggregation* (MSEA), which integrates multi-scale visual attributions to produce more faithful explanations with richer spatial context.

Formally, given an input image $I$, we construct a set of rescaled versions $\{I^{(0)}, I^{(1)}, \ldots, I^{(S)}\}$, where $I^{(0)}$ is the original resolution. The preprocessing depends on the model's input requirements:

$$I^{(s)} = \begin{cases} \text{Resize}(I, \alpha_s), & \text{if the model accepts raw input sizes,} \\ \text{Pad}(\text{Resize}(I, \alpha_s), H, W), & \text{if the model requires fixed input size } (H \times W), \end{cases} \tag{3}$$

where $\alpha_s$ is the scale factor and $\text{Pad}(\cdot, H, W)$ denotes pasting the resized image onto a blank background of size $H \times W$ to preserve positional consistency. For each token $v_i^{(s)}$, its hidden state at the $l^{\text{th}}$ layer is $\mathbf{z}_i^{l,(s)}$, which is projected into the vocabulary space via $\mathbf{W}_U$. To explain a target text token $T_t$, we extract the corresponding logit score $a_i^{l,(s)}(k)$ using equation 2. The attribution map at scale $s$ is then obtained by aggregating the token-level contributions:

$$A^{(s)} = \{a_1^{l,(s)}(k), a_2^{l,(s)}(k), \ldots, a_{N_s}^{l,(s)}(k)\}, \tag{4}$$

where each entry corresponds to the contribution of a visual token to the target text token $T_t$. Finally, to form a holistic explanation, all attribution maps $\{A^{(1)}, \ldots, A^{(S)}\}$ are rescaled to the original image size and fused:

$$A_t = \frac{1}{S} \sum_{s=1}^{S} \text{Resize}\big(A^{(s)}, 1/\alpha_s\big), \tag{5}$$

The aggregated map $A_t$ captures multi-scale contextual information in order to generate explanation for text token $T_t$ with enhanced attribution fidelity.

### 3.3 ACTIVATION RANKING CORRELATION (ARC)

Although logit lens nostalgebraist (2020) provides meaningful visual attribution, the generated maps can be noisy. Previous studies Li et al. (2025a) have shown that this is often caused by interference from residual activations of preceding text tokens. Due to the auto-regressive nature of MLLMs, the prediction of the next token depends on previously generated tokens. Consequently, the hidden state $\mathbf{h}_t^L$ not only encodes information relevant to the current token $T_t$ but also inherits activations from preceding tokens. As demonstrated in Figure 1b, we observe that this effect is particularly pronounced when explaining non-semantic tokens, such as punctuation. For such tokens, decoding visual representations via logit lens yields similar logit scores across many visual patches, resulting in diffuse and noisy attribution maps. In contrast, semantic tokens typically produce high activation only in the relevant visual regions. Motivated by this observation, we propose a method to identify irrelevant preceding tokens and mitigate their impact on the current token's activation.

We introduce *Activation Ranking Correlation* (ARC), a method designed to identify and suppress spurious contextual token activations based on the alignment of top-$k$ predicted token rankings. Unlike prior works Li et al. (2025a) that rely solely on raw logit scores to measure token relevance, ARC evaluates the consistency of the ranking order among top-$k$ predictions. For example, in a driving scenario, the tokens "traffic" and "light" are highly related. When generating "traffic," the logit score for "light" may be low, which could be misinterpreted as low relevance. However, "light" may still appear in the top-$k$ candidates. Using ranking alignment addresses such bias caused by score magnitude.

Formally, for the target text token $T_t$ and its preceding tokens $T_{<t}$ (including prompt and previously generated tokens), we compute the output logits for each token using Equation 1:

$$\mathbf{y}_j^L = \text{softmax}(\mathbf{W}_U \mathbf{h}_j^L), \quad j \leq t. \tag{6}$$

We then extract the indices of the top-$k$ predicted tokens to obtain $\text{rank}_k(\mathbf{y}_j^L)$. The semantic relevance of a preceding token $T_j$ to $T_t$ is quantified using the Rank-Biased Overlap (RBO) Webber et al. (2010) metric $\rho(\cdot, \cdot)$ between their top-$k$ rankings:

$$r_j = \rho\big(\text{rank}_k(\mathbf{y}_j^L), \text{rank}_k(\mathbf{y}_t^L)\big). \tag{7}$$

Tokens with high $r_j$ values are considered semantically coherent with $T_t$, while those with low or negative $r_j$ are treated as irrelevant. To suppress interference, visual attribution maps $A_j$ for each preceding token are computed using Equation 5. We then compute the aggregated irrelevant attribution for $T_t$ as follows:

$$\hat{\mathbf{A}}_t = \frac{1}{\sum_{j<t}(1-r_j)} \sum_{j<t}(1-r_j)\mathbf{A}_j, \tag{8}$$

where $(1-r_j)$ assigns higher weights to spurious tokens. To unify this computation, we additionally define a base attribution $A_0$, derived as the raw attribution (Equation 4) for the text token with

| Method | Type | COCO Caption | | | GranDf | | | OpenPSG | | |
|---|---|---|---|---|---|---|---|---|---|---|
| | | Obj-IoU | Func-IoU | **F1-IoU** | Obj-IoU | Func-IoU | **F1-IoU** | Obj-IoU | Func-IoU | **F1-IoU** |
| Grad-CAM 2017 | Gradient | 21.23 | 51.93 | 30.14 | 17.85 | 62.15 | 27.74 | 22.93 | 48.57 | 31.15 |
| Grad-CAM++ 2018 | | 19.52 | 62.83 | 29.78 | 17.30 | 73.42 | 28.01 | 22.21 | 59.95 | 32.41 |
| Grad-Rollout 2020 | | 1.27 | 99.51 | 2.51 | 1.40 | 99.61 | 2.77 | 1.57 | 99.58 | 13.08 |
| Layer-CAM 2021 | | 11.43 | 84.88 | 20.15 | 13.11 | 82.09 | 22.62 | 14.12 | 85.29 | 24.22 |
| Attention | Attention | 8.20 | 92.87 | 15.07 | 9.60 | 93.56 | 17.42 | 10.58 | 94.28 | 19.03 |
| Attention-Rollout 2020 | | 5.74 | 96.50 | 10.83 | 7.21 | 96.65 | 13.42 | 7.94 | 97.04 | 14.68 |
| CP-LRP 2022 | Combination | 9.90 | 53.97 | 16.73 | 12.61 | 53.24 | 20.39 | 13.30 | 53.36 | 21.30 |
| Attn-LRP 2024 | | 9.92 | 52.41 | 16.69 | 12.15 | 52.19 | 19.72 | 12.78 | 52.26 | 20.54 |
| CAM 2016 | Logit | 21.23 | 51.93 | 30.14 | 17.85 | 62.15 | 27.74 | 22.93 | 48.57 | 31.15 |
| Archi.-Surgery 2025b | | 15.69 | 63.82 | 25.19 | 16.59 | 62.28 | 26.20 | 19.83 | 58.77 | 29.65 |
| TAM Li et al. (2025a) | | 27.37 | 68.44 | 39.10 | 18.65 | 88.97 | 30.83 | 26.26 | 92.99 | 40.95 |
| **Ours** | | 29.35 | 91.57 | **44.45** | 23.32 | 91.85 | **37.20** | 29.20 | 94.69 | **44.64** |

Table 1: Comparison with state-of-the-art methods using Qwen2-VL-2B across diverse datasets. Our method consistently yields superior visual explanations, as evidenced by the highest F1-IoU scores across all evaluated datasets.

minimal visual activation, i.e., the token with index $k = \arg\min_k \sum_1^{N_s} a_1^l$. The refined attribution for $T_t$ is then obtained by subtracting the scaled irrelevant attribution:

$$\tilde{\mathbf{A}}_t = \mathcal{G}(\lfloor \mathbf{A}_t - \beta\hat{\mathbf{A}}_t \rfloor_+), \quad \beta = \arg\min_\beta ||\mathbf{A}_t - \beta\hat{\mathbf{A}}_t||_2^2, \tag{9}$$

where $\lfloor \cdot \rfloor_+$ retains only positive activations. Following prior work Li et al. (2025a), we use an adaptive scaling factor $\beta$ to control the suppression strength to avoid over-mitigation and a Rank Gaussian Filter $\mathcal{G}$ as post-processing to further reduce noise. The resulting refined attribution $\tilde{\mathbf{A}}_t$ effectively diminishes contamination from irrelevant contexts and enhances the fidelity of explanations.

Together, MSEA and ARC provide complementary benefits: MSEA improves visual attribution by incorporating multi-scale spatial context, while ARC mitigates noisy activations from preceding tokens, jointly producing more faithful explanations.

## 4 EXPERIMENTS

### 4.1 EXPERIMENTAL SETUP

**Models and Datasets.** We evaluate our approach on three representative multimodal large language models (MLLMs) with distinct architectures: LLaVA-1.5 Liu et al. (2024), Qwen2-VL Wang et al. (2024), and InternVL2.5 Chen et al. (2024). To assess scalability, we further experiment with model variants spanning 2B to 13B parameters (see Table 2). Our evaluation relies on datasets that provide both textual descriptions and pixel-level segmentation masks. The primary benchmark is the COCO Caption dataset Chen et al. (2015), which uses images and annotations from COCO 2014 Lin et al. (2014). Since our method is purely post-hoc and requires no training, we evaluate exclusively on the standard 5K-image minival split. We additionally include GranDf Rasheed et al. (2024) (1K images) and the validation set of OpenPSG Zhou et al. (2024) (3,176 images). Ground-truth masks in COCO Caption dataset and GranDf dataset are human-annotated, whereas those in OpenPSG are derived from the integrated annotations provided by Rasheed et al. Rasheed et al. (2024).

**Implementation Details.** For hyperparameters, the default scale factors in Equation 3 are set to $[0.5, 0.75, 1.0]$, with additional configurations evaluated as shown in Figure 3. Logit lens nostalgebraist (2020) is applied on the hidden states from the last transformer layer (Equation 4). For ranking-based correlation, we set $k = 50$ to select the top-$k$ logits indices (Equation 7). For evaluation, we adopt three metrics following prior work: Obj-IoU, Func-IoU, and F1-IoU. Obj-IoU measures the intersection-over-union between generated attribution maps and ground-truth masks. Func-IoU quantifies activations on non-semantic tokens (e.g., punctuation), where higher values indicate fewer false positives. To jointly account for both Obj-IoU and Func-IoU, F1-IoU is reported as the primary metric, defined as follows:

$$\text{F1-IoU} = \frac{2 \cdot \text{Obj-IoU} \cdot \text{Func-IoU}}{\text{Obj-IoU} + \text{Func-IoU}}. \tag{10}$$

| Method | MLLM | COCO Caption | | | GranDf | | | OpenPSG | | |
|--------|------|---------|----------|--------|---------|----------|--------|---------|----------|--------|
| | | Obj-IoU | Func-IoU | F1-IoU | Obj-IoU | Func-IoU | F1-IoU | Obj-IoU | Func-IoU | F1-IoU |
| CAM | | 23.17 | 43.16 | 30.16 | 20.07 | 47.48 | 28.21 | 25.11 | 51.55 | 33.77 |
| TAM | LLaVA1.5-7B | 27.65 | 61.43 | 38.13 | 20.71 | 59.15 | 30.68 | 28.57 | 61.06 | 38.93 |
| **Ours** | | 30.62 | 87.32 | 45.34 | 24.79 | 85.18 | 38.40 | 32.03 | 86.80 | 46.79 |
| CAM | | 24.82 | 51.18 | 33.43 | 21.34 | 43.99 | 28.74 | 26.65 | 48.45 | 34.39 |
| TAM | LLaVA1.5-13B | 29.12 | 58.50 | 38.88 | 22.10 | 51.02 | 30.84 | 30.88 | 59.96 | 40.76 |
| **Ours** | | 31.76 | 97.18 | 47.87 | 26.08 | 95.58 | 40.98 | 32.57 | 97.32 | 48.80 |
| CAM | | 15.94 | 45.62 | 23.63 | 18.28 | 37.64 | 24.61 | 19.76 | 46.42 | 27.72 |
| TAM | InternVL2.5-2B | 21.38 | 65.10 | 32.19 | 20.48 | 85.93 | 33.08 | 23.00 | 86.86 | 36.36 |
| **Ours** | | 30.61 | 76.48 | 43.72 | 24.54 | 88.93 | 38.47 | 31.50 | 91.03 | 46.81 |
| CAM | | 18.23 | 40.95 | 25.23 | 20.91 | 44.52 | 28.46 | 21.28 | 34.70 | 26.38 |
| TAM | InternVL2.5-4B | 21.76 | 63.12 | 32.36 | 22.53 | 89.71 | 36.02 | 23.49 | 89.75 | 37.23 |
| **Ours** | | 31.80 | 82.73 | 45.94 | 27.73 | 94.34 | 42.86 | 33.52 | 94.09 | 49.43 |
| CAM | | 14.59 | 64.41 | 23.80 | 18.04 | 57.42 | 27.45 | 18.46 | 62.21 | 28.47 |
| TAM | InternVL2.5-8B | 19.98 | 66.53 | 30.73 | 21.56 | 85.95 | 34.47 | 21.73 | 88.74 | 34.91 |
| **Ours** | | 32.16 | 73.20 | 45.25 | 27.00 | 86.72 | 41.18 | 33.97 | 91.11 | 49.49 |
| CAM | | 21.23 | 51.93 | 30.14 | 17.85 | 62.15 | 27.74 | 22.93 | 48.50 | 31.15 |
| TAM | Qwen2-VL-2B | 27.37 | 68.44 | 39.10 | 18.65 | 88.97 | 30.83 | 26.26 | 92.99 | 40.95 |
| **Ours** | | 29.35 | 91.57 | 44.45 | 23.32 | 91.85 | 37.20 | 29.20 | 94.69 | 44.64 |
| CAM | | 22.51 | 42.44 | 29.42 | 18.60 | 68.03 | 29.21 | 23.41 | 42.94 | 30.30 |
| TAM | Qwen2-VL-7B | 28.13 | 71.85 | 40.43 | 19.88 | 90.57 | 32.61 | 26.94 | 89.88 | 41.45 |
| **Ours** | | 29.86 | 94.77 | 45.41 | 23.53 | 90.59 | 37.35 | 29.01 | 94.33 | 44.37 |

Table 2: Comparison with state-of-the-art methods across MLLMs of diverse architectures and parameter scales. Our approach demonstrates consistently better performance, highlighting its broad compatibility with different model architectures and robust scalability across varying model sizes.

## 4.2 QUANTITATIVE RESULTS

**Comparison with SoTA Methods.** We evaluate our method against a comprehensive suite of state-of-the-art visual explanation approaches, including gradient-based, attention-based, hybrid (gradient-attention), and logit-based methods, across three benchmark datasets: COCO Caption dataset Chen et al. (2015), GranDf dataset Rasheed et al. (2024), and OpenPSG dataset Zhou et al. (2024). As shown in Table 1, our approach consistently achieves the highest Obj-IoU and overall F1-IoU scores across all datasets and across all categories of explanation methods, indicating more precise and faithful visual attributions. Compared to the current state-of-the-art method, TAM, our method yields absolute improvements of 5.35%, 6.37%, and 3.69% in F1-IoU on COCO Caption, GranDf, and OpenPSG dataset, respectively. Notably, on COCO Caption dataset, this gain is primarily driven by a 23.13% improvement in Func-IoU, reflecting a substantial reduction in false positives and a stronger ability to suppress interference from activations of preceding tokens.

**Generalization across MLLM Architectures and Scales.** To evaluate the generalizability of our approach, we conduct an extensive comparison across seven multimodal large language models (MLLMs), spanning three representative architectures—LLaVA-1.5, InternVL2.5, and Qwen2-VL—and parameter scales from 2B to 13B. As shown in Table 2, our method consistently outperforms state-of-the-art baselines (CAM Zhou et al. (2016) and TAM Li et al. (2025a)) across all models and datasets. On the COCO Caption dataset, we observe particularly strong gains: our approach surpasses TAM by 8.99% F1-IoU on LLaVA-1.5-13B and by 13.58% on InternVL2.5-8B. Notably, within the InternVL2.5 family, performance improvements grow with model scale—yielding 11.53%, 13.58%, and 14.52% absolute gains in F1-IoU for the 2B, 4B, and 8B variants, respectively. This positive scaling trend suggests that our method not only generalizes across diverse architectures but also benefits from increased model capacity, underscoring its better scalability.

**Ablation Study.** We conduct ablation studies on the COCO Caption dataset to evaluate the effectiveness of our proposed components: Multi-Scale Explanation Aggregation (MSEA) and Activation Ranking Correlation (ARC). For ARC, we also compare with another interference mitigation strategy TAM. The results are summarized in Table 3. First, the introduction of ARC leads to a substantial improvement in Func-IoU, indicating a significant reduction in false positives. Specifically, on Qwen2-VL-2B, adding ARC yields a 48.09% absolute gain in Func-IoU. Even when

| MSEA | | Interference Mitigation | | | Qwen2-VL-2B | | | LLaVA1.5-7B | | |
|------|------|------|------|------|---------|----------|--------|---------|----------|--------|
| Mean | Max | **ARC** | TAM | Mean | Obj-IoU | Func-IoU | **F1-IoU** | Obj-IoU | Func-IoU | **F1-IoU** |
| | | | | | 24.82 | 43.34 | 31.57 | 25.41 | 39.74 | 31.00 |
| | | | | ✔ | 27.84 | 49.85 | 35.72 | 27.65 | 61.43 | 38.13 |
| | | | ✔ | | 27.37 | 68.44 | 39.10 | 27.81 | 66.22 | 39.17 |
| | | ✔ | | | 27.07 | 91.43 | $41.78_{+10.21}$ | 28.04 | 85.15 | $42.19_{+11.19}$ |
| | ✔ | | | | 24.32 | 63.77 | 35.21 | 27.49 | 52.53 | 36.09 |
| ✔ | | | | | 25.81 | 63.21 | $36.65_{+5.08}$ | 27.59 | 48.88 | $35.27_{+4.27}$ |
| | ✔ | ✔ | | | 27.77 | 91.64 | 42.63 | 30.12 | 85.32 | 44.52 |
| ✔ | | ✔ | | | 29.35 | 91.57 | $44.45_{+12.88}$ | 30.62 | 87.32 | $45.34_{+14.34}$ |

Table 3: Ablation study on the COCO Caption dataset Chen et al. (2015) dataset using Qwen2-VL-2B Wang et al. (2024). ECI denotes Estimated Causal Inference. "RelSort" and "Mean" are components within ECI, and "TAM" is our proposed module. The combination of modules is mutually beneficial — the gain exceeds the sum of individual improvements. Metrics are IoU for object words, IoU for function words, and their F1-score-like combination, respectively.

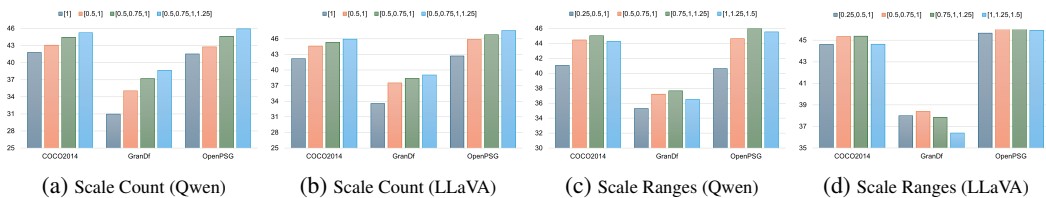

| (a) Scale Count (Qwen) | (b) Scale Count (LLaVA) | (c) Scale Ranges (Qwen) | (d) Scale Ranges (LLaVA) |
|---|---|---|---|

Figure 3: Performance sensitivity to the number and range of scaling factors across datasets and model architectures. Subfigures (a) and (b) show the impact of varying the number of scaling factors, while subfigures (c) and (d) illustrate the effect of different ranges of scaling factors.

compared against the strong baseline TAM, our method still achieves a 22.99% improvement. Similar trends are observed on LLaVA-1.5-7B, confirming ARC's consistent ability to suppress spurious influences from irrelevant preceding tokens. For MSEA, both mean and max aggregation strategies improve F1-IoU over the baseline, demonstrating the benefit of incorporating multi-scale visual context. More importantly, MSEA and ARC exhibit strong complementary effects: when combined, they further boost attribution performance beyond what either component achieves alone. Specifically, the joint use of MSEA and ARC improves F1-IoU by 2.67% on Qwen2-VL-2B and 3.15% on LLaVA-1.5-7B compared to using ARC alone, validating that modeling intra-modal interactions in both vision and language is essential for faithful multimodal explanations.

**Sensitivity Analysis on Scale Factors.** Figure 3 investigates the sensitivity of our MSEA module to the number and range of scaling factors used during multi-scale attribution computation, evaluated across two model architectures (Qwen2-VL and LLaVA-1.5) and three datasets (COCO2014, GranDf, and OpenPSG). Subfigures (a) and (b) reveal a consistent trend: increasing the number of scales from 1 to 4 significantly improves attribution performance, suggesting that richer spatial context enhances explanation fidelity. Notably, using four scales, e.g., $[0.5, 0.75, 1.0, 1.25]$, yields peak performance across both models, underscoring the importance of multi-scale aggregation in visual attribution. Subfigures (c) and (d) further analyze the impact of different scale ranges. We observe that moderate ranges, such as $[0.5, 0.75, 1]$ or $[0.75, 1.0, 1.25]$, consistently outperform both overly narrow (e.g., $[0.25, 0.5, 1.0]$) and excessively large ranges (e.g., $[1.0, 1.25, 1.5]$). This indicates that carefully selected rescaled resolutions better capture contextual cues critical for faithful attribution. Moreover, LLaVA-1.5 exhibits lower sensitivity to scale range compared to Qwen2-VL. We attribute this difference to their distinct image preprocessing strategies: Qwen2-VL accepts raw image sizes and is thus more affected by rescaling, whereas LLaVA-1.5 uses fixed-resolution inputs.

### 4.3 VISUALIZATION

We visualize the attribution maps generated by the Qwen2-VL-2B model (Figure 4) and the LLaVA-1.5-7B model (Figure 5) on the COCO Caption dataset. The results show that our method produces more holistic and faithful attributions compared to the state-of-the-art TAM method. Specifically,

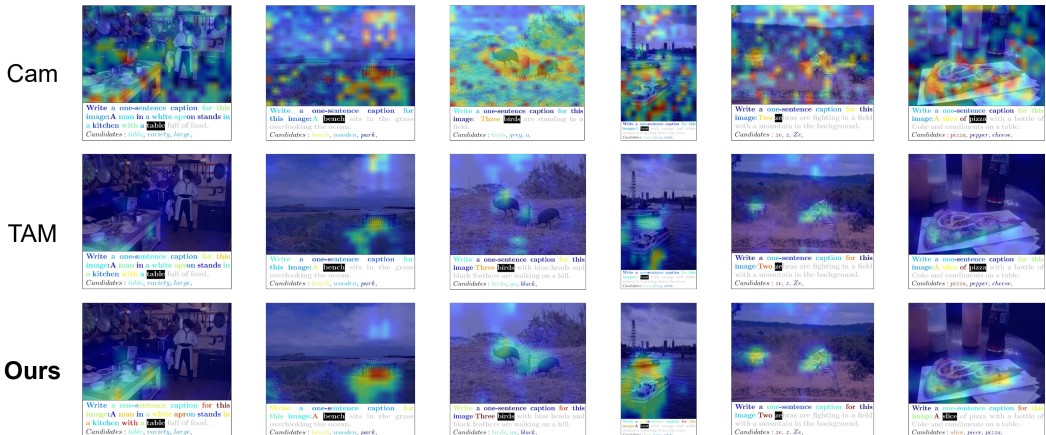

Figure 4: Visualization of attribution maps generated using the Qwen2-VL-2B model.

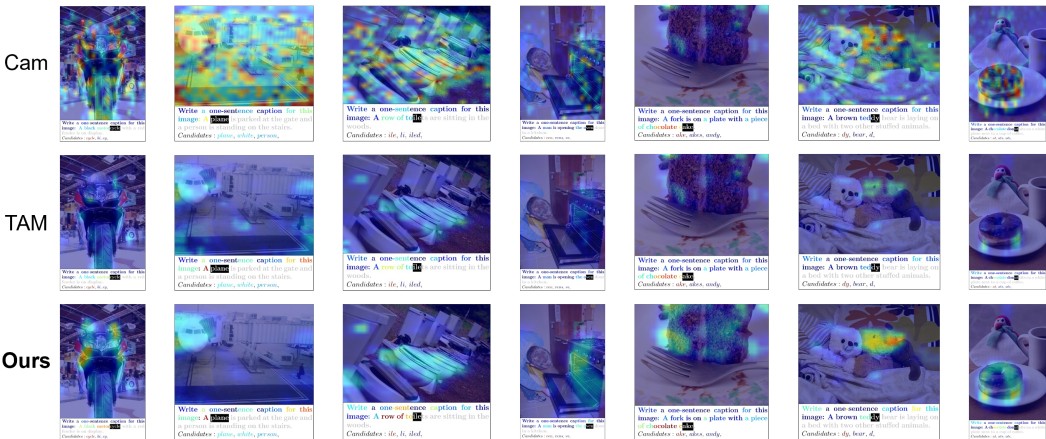

Figure 5: Visualization of attribution maps generated using the LLaVA-1.5-7B model.

our attributions exhibit stronger alignment with the target objects and significantly less noise in semantically irrelevant regions.

## 5 CONCLUSION

In this work, we addressed the critical challenge of enhancing the interpretability of Multimodal Large Language Models (MLLMs) by tackling a key limitation in existing methods: their neglect of intra-modal interactions. We argued that a holistic understanding of MLLM reasoning requires more than just cross-modal attribution. To this end, we introduced a new explanation method featuring two complementary components: Multi-Scale Explanation Aggregation (MSEA) and Activation Ranking Correlation (ARC). Our research demonstrates that by integrating intra-modal interactions, we can produce significantly more faithful and robust explanations. MSEA leverages information from multiple image scales to capture crucial spatial context among visual tokens, while ARC effectively mitigates noise by identifying and suppressing the influence of irrelevant preceding text tokens. Our comprehensive empirical evaluations on a variety of state-of-the-art MLLMs, including LLaVA-1.5, Qwen2-VL, and InternVL2.5, and across diverse benchmarks, consistently showed that our approach outperforms existing explainability methods. We achieved quantitative improvements ranging from 3.69 to 14.52% across different models and tasks, validating the effectiveness and generalizability of our framework.

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

## A APPENDIX

### A.1 USE OF LLM

Declaration: The use of Qwen in the preparation of this manuscript was strictly limited to grammatical correction and text polishing.

