# OpenReview forum: "Explaining Multimodal LLMs via Intra-Modal Token Interactions"
_ICLR.cc/2026/Conference — ICLR 2026 Conference Withdrawn Submission_

### Official Review · Reviewer_3Q2o · 2025-10-27

**Soundness:** 1
**Presentation:** 2
**Contribution:** 1
**Rating:** 2
**Confidence:** 3

**Summary:**

The paper proposes MSEA (Multi-Scale Explanation Aggregation) and ARC (Activation Ranking Correlation) to produce token-level visual attributions for multimodal LLMs. MSEA averages logit-lens maps computed from multiple input resolutions to reduce spatial fragmentation; ARC down-weights interference from preceding text tokens using rank-based similarity between their next-token distributions and that of the target token. Evaluations on captioning/scene datasets report IoU-style improvements over baselines (e.g., TAM).

**Strengths:**

1. Clear, post-hoc, model-agnostic recipe (no retraining); easy to plug into popular MLLMs.

2. Empirical gains on localization-type metrics across several models/datasets.

**Weaknesses:**

1. Overstated novelty about “intra-modal interactions.”
Full-graph methods (gradients, LRP/AttnLRP, attention rollout, TAM/LLaVA-CAM) already propagate relevance through all cross- and intra-modal paths; interaction effects are not “ignored” by default.

2. Receptive-field misconception.
In ViTs, global self-attention makes a token’s effective receptive field image-wide after early layers. Fragmented maps arise from attribution noise/aggregation choices, not from an inherently local RF. MSEA’s value is ensembling across tokenizations, not “expanding RF.”

3. Baselines lack crucial details.
Which layer/heads for CAM/Grad-CAM? How was AttnLRP configured? Without this, fairness and reproducibility are uncertain. The authors should write details at supplementary material.

4. No faithfulness evaluation.
Results rely on human-grounded IoU and “cleanliness.” There are no causal tests (insertion/deletion curves, AOPC/ROAR, evidence-erasure, logit drop when masking top-k regions). MSEA/ARC may yield prettier maps yet be less causal.

5. Qualitative analysis is thin.
Few examples; captions are terse; comparisons focus mainly on TAM. Lacks side-by-sides against other listed baselines (Grad-CAM/++, Rollout, LRP/AttnLRP, IG-style).

6. Cost/latency not discussed.
multi-scale forward passes + ARC scoring add overhead; no wall-clock or memory analysis vs TAM/AttnLRP/perturbation methods.

7. There is no detailed captions at every figure.

**Questions:**

See above weaknesses.

---

### Official Review · Reviewer_SxV5 · 2025-10-27

**Soundness:** 2
**Presentation:** 3
**Contribution:** 2
**Rating:** 4
**Confidence:** 3

**Summary:**

This paper proposes a novel framework to enhance the explainability (attribution) of Multimodal Large Language Models (MLLMs), focusing specifically on Intra-Modal Interactions. The framework aims to overcome the issues of fragmented, incoherent, and noise-contaminated explanations prevalent in existing methods, striving to generate more Faithful and Fine-grained attributions. The core contributions include:

1. Multi-Scale Explanation Aggregation (MSEA): For the visual modality, this aggregates attribution results from different input scales, dynamically adjusting the receptive fields of visual tokens to capture spatial context, leading to more holistic visual explanations.

2. Activation Rank Correlation (ARC): For the text modality, this method quantifies semantic relevance by comparing the Top-k prediction rank consistency (using the RBO metric) between the current token and context tokens. This mechanism is used to suppress spurious activations and noise originating from irrelevant contexts.

The approach is post-hoc and training-free. It is validated across state-of-the-art MLLMs like LLaVA-1.5 and Qwen2-VL, showing consistent superiority over existing baselines (e.g., TAM) in metrics like Obj-IoU, Func-IoU, and F1-IoU.

**Strengths:**

1. Strong Solution to Technical Problems: The mechanistic design of MSEA and ARC effectively addresses known limitations in Logit Lens attribution. The introduction of Top-k Prediction Rank Correlation (RBO) in the text modality is a clever and effective mechanism for filtering semantic noise.

2. Extensive Experimental Validation: Comprehensive testing across representative MLLM architectures (LLaVA-1.5, Qwen2-VL, and InternVL2.5) increases the credibility of the results.

3. Practicality: As a post-hoc method, it requires no retraining or fine-tuning of the MLLM, making it easy to deploy and apply.

**Weaknesses:**

1. Causal Logic Risk (MSEA): Multi-Scale Explanation Aggregation (MSEA) obtains attribution by altering the input scale, which may introduce a causal logic inconsistency. The attribution result may not strictly reflect the model's decision under the original input, and this lacks rigorous theoretical or empirical support.

2. Missing Diagnostic Application/Value: Although the method achieves high fidelity, the paper lacks diagnostic case studies focusing on real model errors like MLLM Hallucination. Analyzing such error cases is crucial to demonstrate the method's diagnostic scope and application value, enabling attribution to truly fulfill its role in model analysis.

3. Computational Burden: Since both MSEA (involving multiple forward passes for different scales) and ARC (involving Top-k rank consistency calculation, which can be resource-intensive with long contexts/outputs) are added on top of the base Logit Lens method, the computational overhead may be high, especially when applied to very long contexts or complex MLLMs.

**Questions:**

1. Causal Consistency (MSEA): Given that MSEA aggregates attribution results by altering the input scale, we are concerned about potential causal logic inconsistencies. Could the authors provide deeper theoretical arguments or additional experimental evidence (e.g., ensuring the original decision remains unchanged) to guarantee the fidelity of the MSEA attribution results with respect to the model's decision mechanism under the original input?

2. Practical Application (Hallucination Diagnosis): Since the method aims to improve attribution fidelity, we suggest the authors provide qualitative case analyses demonstrating how MSEA and ARC can effectively diagnose the root causes of MLLM Hallucinations (whether due to visual signal misinterpretation or spurious activations from text context). A comparison with baseline methods would be highly beneficial.

3. RBO Sensitivity: Regarding the ARC module, we encourage the authors provide ablation studies or sensitivity analysis concerning the choice of the Top-k value (e.g., the default $k=50$), to prove the stability and reasonableness of this parameter selection.

4. Computational Complexity: Considering the overhead of MSEA (multiple forward passes) and ARC's RBO calculation, could the authors provide a practical time breakdown? Specifically, what is the required wall-clock time to analyze a single case for different context lengths?

---

### Official Review · Reviewer_hFsC · 2025-10-28

**Soundness:** 3
**Presentation:** 3
**Contribution:** 3
**Rating:** 6
**Confidence:** 4

**Summary:**

This paper addresses the limitation of current interpretability studies on Multimodal Large Language Models (MLLMs), which mainly focus on cross-modal attribution while neglecting intra-modal dependencies. The authors point out that, within the visual modality, existing methods often divide images into independent patches and thereby overlook the importance of spatial contextual relationships. To address this issue, the paper proposes a Multi-Scale Explanation Aggregation (MSEA) approach, which aggregates attributions across multiple image resolutions to dynamically adjust the receptive field, resulting in more coherent visual explanations.
Within the textual modality, to mitigate spurious activations caused by preceding tokens, the paper introduces an Activation Ranking Correlation (ARC) method. By evaluating the ranking consistency among the Top-k predicted tokens, ARC quantifies the relevance between each contextual token and the current token, and further suppresses spurious activations from irrelevant contexts based on this correlation.
Experiments conducted on multiple MLLMs and datasets demonstrate that the proposed method achieves superior interpretability quality compared with existing mainstream approaches.

**Strengths:**

The paper innovatively focuses on the intra-modal dependencies in the interpretability of Multimodal Large Language Models (MLLMs). In the visual modality, it captures more contextual visual information by adjusting the visual receptive field. In the textual modality, it suppresses irrelevant spurious activations by quantifying the ranking consistency among Top-k predicted tokens. Based on the experimental results and visualizations provided by the authors, the proposed method demonstrates superior performance compared to existing interpretability approaches.

**Weaknesses:**

The presentation and explanation of the proposed method are overly abstract, with some steps omitted and theoretical aspects insufficiently detailed, which may make it difficult for readers to fully understand. Furthermore, some of the visualized results and analyses are limited.

(1) The explanation of “logits” and the “Logit Lens interpretability mechanism” in Section 3.1 is somewhat abstract, which may make it difficult for readers to clearly understand the related background.
(2) The motivation and interpretation of the proposed input–output level operations in Section 3.2 are also rather abstract, which may affect readers’ comprehension.
(3) It is recommended to clarify in Section 3.2 which variable remains constant when the image scale is changed, and how this affects the variation of the receptive field.
(4) As shown in Figure 2, it appears that one patch is divided into four sub-patches after tokenization. It is suggested to provide a clearer explanation of this process or include a formula to describe it.
(5) For Equation (5), it is recommended to provide a more detailed explanation of how the results are rescaled to the original image size after the aggregation in Equation (4).
(6) It is recommended to provide a more detailed explanation of the use of Rank-Biased Overlap (RBO) in Equation (7) and the settings of its parameters.
(7) In Section 3.3, the definition of “base attribution” is not clearly described, particularly regarding its computation for the token with index k. Moreover, the way in which the defined “base attribution” is utilized is not clearly explained.
(8) Figure 2 does not clearly illustrate the proposed method, especially the depiction of ARC, which appears misaligned with the textual description. It is recommended to refine both the figure and the corresponding explanation.
(9) It is recommended to provide a detailed explanation of each evaluation metric, including its computation method and meaning, to help readers better understand these metrics.
(10) It is suggested to provide additional visualizations of non-semantic tokens.

**Questions:**

see weakness.

---

### Official Review · Reviewer_UEeF · 2025-11-01

**Soundness:** 2
**Presentation:** 2
**Contribution:** 2
**Rating:** 4
**Confidence:** 4

**Summary:**

This paper proposes a framework to enhance the interpretability of Multimodal Large Language Models (MLLMs) by intra-modal interactions in vision-language modalities. The two core contributions are: 1. Multi-Scale Explanation Aggregation (MSEA): Aggregates visual token attributions across multiple image resolutions; 2. Activation Ranking Correlation (ARC): Identifies and suppresses spurious activations from irrelevant preceding tokens in the textual modality, based on alignment of top-k token ranking predictions.

**Strengths:**

1. MSEA and ARC complementary work together to improve both visual and textual interpretability. SEA captures spatial context across visual tokens, while ARC reduces noise in textual attributions caused by irrelevant context tokens.

2. Quantitative improvements and qualitative visualization results support the effectiveness of the approach.

3. Good writing and presentation.

**Weaknesses:**

1. High Overlap with TAM and Incremental Innovations:

The paper’s methodology is built upon TAM[1]. MSEA is a straightforward extension of multi-scale input techniques commonly used in computer vision tasks, offering limited innovation. SEA is also a technique reuse on LLM inference.

2. Computational Complexity:

MSEA introduces significant computational overhead due to multi-resolution processing and attribution aggregation, a clear trade-off on significant increased complexity and performance.

ARC share a similar case, that requires token-ranking alignment computations for every preceding token, further compounding the computational cost.

The paper does not provide a detailed analysis of runtime performance or scalability trade-offs, which raises concerns about practical applicability.

**Questions:**

1. What is the runtime cost of MSEA and ARC compared to TAM or other baseline methods?

2. How does the method scale with larger models?

3. Robustness: How does the method handle challenging scenarios, such as: Images with multiple similar objects (e.g., distinguishing specific birds or cars)? Occluded objects where parts of the image are missing? Interaction-heavy scenes involving relational reasoning (e.g., a person interacting with tools)?

---

### Note · Authors · 2025-11-17

I have read and agree with the venue's withdrawal policy on behalf of myself and my co-authors.